# Unraveling Investor Behavior: The Role of Hyperbolic Discounting in Panic Selling Behavior on the Global COVID-19 Financial Crisis

**DOI:** 10.3390/bs14090795

**Published:** 2024-09-09

**Authors:** Sumeet Lal, Trinh Xuan Thi Nguyen, Aliyu Ali Bawalle, Mostafa Saidur Rahim Khan, Yoshihiko Kadoya

**Affiliations:** School of Economics, Hiroshima University, 1-2-1 Kagamiyama, Higashihiroshima 7398525, Japan; nguyen93@hiroshima-u.ac.jp (T.X.T.N.); d231201@hiroshima-u.ac.jp (A.A.B.); khan@hiroshima-u.ac.jp (M.S.R.K.); ykadoya@hiroshima-u.ac.jp (Y.K.)

**Keywords:** hyperbolic discounting, panic selling, COVID-19, Japan

## Abstract

In financial markets, irrational behaviors such as hyperbolic discounting and panic selling are prevalent. However, their widespread empirical associations remain unexplored. Numerous behavioral theories discuss how cognitive biases exacerbate panic selling through the lens of immediate loss aversion, a phenomenon in which individuals exhibit impulsive decision-making tendencies due to an intense fear of financial loss during market upheaval. Despite the theoretical elucidation, empirical investigations of these dynamics are lacking. Using a robust dataset comprising 121,293 active investors sourced from a collaborative effort between Hiroshima University and Rakuten Securities Inc., this study used mean comparison tests and probit regression to analyze hyperbolic discounting’s role in panic selling behavior on the global COVID-19 financial crisis. The findings reveal that hyperbolic discounting plays a central role in triggering investors’ impulsive panic selling behavior, which is driven primarily by fear of potential losses. Other factors that influence panic selling behavior include age, male gender, low education level, financial literacy, household income, household assets, risk aversion, and overconfidence in financial knowledge. Our study explicates the need to address cognitive biases in financial decision making during market crises through strategies such as targeted financial education, regulatory interventions against market manipulation, and the provision of professional advice to investors.

## 1. Introduction

In financial markets, despite the prevalence of irrational behaviors such as hyperbolic discounting and panic selling, their global empirical associations remain unexplored. Panic selling, characterized by the hasty divestment of stocks and funds, emerges as a natural response to turbulent and uncertain market conditions [1,2]. During such crises, the fear of loss intensifies due to the decreasing value of stocks [3], causing individuals to act impulsively in the financial market. This impulsivity, a key aspect of hyperbolic discounting [4], leads individuals to prioritize immediate relief from emotional and financial distress over rational, long-term profitability by divesting from stocks [5]. Furthermore, behavioral economics theories [6,7] provide additional justification for the intertwined nature of panic selling and hyperbolic discounting arising from unforeseen circumstances. These theories elucidate how cognitive biases such as loss aversion and present bias influence individual financial decision-making processes without considering the potential consequences of market crises. Building on established theories of behavioral economics, we hypothesize that the stronger the hyperbolic discounting tendency, the greater the likelihood of engaging in panic selling during market downturns. We argue that, as discussed in the literature, hyperbolic discounting creates an overreaction to information and motivates people to take risky positions with higher expected short-term returns [8]; it also leads people to overreact to negative information, placing greater emphasis on loss and regret aversion and motivating investors to sell stocks. Nevertheless, the absence of empirical research on the association between hyperbolic discounting and panic selling leaves a critical gap in the understanding of investor behavior during market crises.

Panic selling arises from the tendency of panic sellers to act impulsively due to fear and emotional distress, causing suboptimal outcomes such as realizing losses and foregoing long-term profits [9]. This behavior contradicts the long-term investment principles advocated by Siegel [10], who emphasizes the historical resilience of equity markets over time (Figure 1) and that the retention of stocks during uncertainty effects increased value over time [11].

Moreover, panic selling challenges the efficient market hypothesis and raises questions about the random walk theory, both of which traditionally assume rational behavior among investors [12]. According to the efficient market hypothesis, irrational behavior such as panic selling should be quickly corrected by rational investors who recognize mispricing and act accordingly [12]. However, widespread panic selling suggests that market prices can deviate significantly and persistently from their fundamental values, challenging the notion of always-efficient markets. While the random walk theory posits that stock prices should move randomly and be unpredictable [12], it is important to acknowledge that selling based on new information, such as the economic impact of the COVID-19 pandemic, may be rational if that information justifies a belief in future declines. Nevertheless, selling large portions of a portfolio in the absence of a clear, rational assessment of the individual’s financial situation or long-term goals may indicate panic selling, particularly when the decision appears to be driven more by market volatility and emotional responses than by a rational analysis of the new information. Several studies indicate that markets often rebound after downturns, suggesting that selling during such periods may not yield long-term benefits (Brown and Cliff [13]; Sauer and Kramer [14]; Vanguard [15]). The recent shock to financial markets induced by the COVID-19 pandemic caused a significant decline in stock prices [16]. However, this downturn lasted only a few months, as shown in Figure 2, after which the value of the stocks began to recover, demonstrating the resilience of the stocks to generate long-term profits [17].

In addition, Zhou [18] and Elkind et al. [19] suggest that people who panic sell their stocks may exit the market and are less likely to reenter the stock market in the future. This limited participation due to fear impedes wealth accumulation in the long run [20]. Therefore, to investigate the detrimental effects of panic selling on long-term profitability, we examine the influence of hyperbolic discounting on investor behavior, which is prevalent particularly during periods of market turmoil.

Theoretically, the association between panic selling and hyperbolic discounting could be established through at least four channels. First, the loss aversion phenomenon of the prospect theory asserts that individuals strongly discount the value of future rewards compared to immediate outcomes [21]. This means that they might prioritize avoiding immediate losses over potential long-term gains. During market turmoil, the immediate pain of seeing a portfolio’s value drop sharply makes investors anxious. Hyperbolic discounting drives them to sell assets to avoid further immediate losses even if it means missing out on potential future recoveries. Second, the theory of present bias asserts that individuals focus on the present often at the expense of future benefits [22,23]. Investors might ignore long-term investment strategies and fundamentals, instead focusing on the immediate market downturn. The fear of short-term losses overshadows the consideration of long-term growth. Third, the theory of intertemporal tradeoff asserts that preferences can change dramatically over time [6,7]. What seemed like a rational long-term plan during stable periods may seem intolerable during market turmoil. Investors might have initially planned to hold their investments for the long term, but as the market drops, their preferences change. The immediate desire to stop further losses overrides the original long-term strategy. Finally, emotional reactions to immediate outcomes can be strong and influence decision-making processes. Fear and anxiety during market declines trigger emotional reactions. Investors may act impulsively, driven by the immediate emotional relief that selling provides, rather than rational analysis. Overall, this theoretical background provides a foundation for understanding the psychological mechanisms driving investor behavior in financial markets and facilitates our pioneering empirical research to explain panic selling from the viewpoint of hyperbolic discounting.

Although our empirical research pioneers in exploring the association between hyperbolic discounting and panic selling, previous studies have investigated the influence of hyperbolic discounting on various other financial decisions. Understanding this body of literature, along with the theoretical underpinnings discussed above, can establish a solid foundation for our hypotheses and guide our predictions. For instance, Love and Pehlan [8] investigate the impact of hyperbolic discounting on asset allocation strategies and find that it affects stock market participation, asset allocation, and savings decisions throughout the life cycle. They observe that hyperbolic discounters tend to accumulate less wealth and participate in the stock market at a later age. When they participate, they tend to hold a higher share of equities, particularly after retirement. Kang and Ye [24] construct a model of corporate investment decisions under hyperbolic discounting and assert that firms with hyperbolic discounting preferences face an underinvestment problem. Jabeen et al. [25] discuss the psychological impact of the COVID-19 pandemic on investor and stock market behavior, suggesting that both rational and irrational responses contribute to a bearish market trend. Therefore, drawing from the consistent link between hyperbolic discounting and irrational financial behavior, this study hypothesizes that hyperbolic discounters engage in panic selling behavior.

This study contributes to the literature in three ways. First, it provides empirical evidence of the role of hyperbolic discounting in influencing panic selling behavior from the perspective of loss aversion and how it causes an impulsive sell-off of stocks and funds. Second, it uses a large-scale, nationally representative dataset of approximately 130,000 active investors from Japan’s leading online security companies, allowing us to conduct a comprehensive analysis of the role of hyperbolic discounting in panic selling. This approach enhances the robustness of our findings and provides a more nuanced understanding of the dynamics during financial instability. Finally, it contributes toward the development of effective financial policies and strategies by policymakers and financial institutions; they can mitigate the impact and promote market stability by understanding the role of hyperbolic discounting and other factors that drive panic selling.

## 2. Data and Methods

### 2.1. Data

This research utilizes a large-scale dataset from the 2023 wave of the “Survey on Life and Money”, which is an online survey conducted by Rakuten Securities in collaboration with Hiroshima University. The survey targeted active account holders of securities companies aged 18 years and above and collected detailed information on the demographic, socioeconomic, and psychological preferences of Japanese adults, focusing on investors’ hyperbolic discounting and panic selling behavior. After conducting the survey, we excluded incomplete responses, which reduced the final sample size to 129,293 observations, representing 68.22% of the original 189,524 valid responses. Incomplete data were excluded when respondents either did not provide answers or provided incomplete data for key variables, including the panic selling and socioeconomic variables (details of the excluded data are not presented here to save on space but are available upon request). Additionally, we compared the distribution patterns of the data before and after the exclusion of incomplete responses and found that the patterns remained largely consistent. This suggests that the exclusion of a portion of the data did not materially affect the nature of the dataset or bias our results. The large sample size and diversity of respondents improve the reliability and validity of the research, providing a solid foundation for exploring the complex dynamics of panic selling behavior and hyperbolic discounting in the context of the Japanese market during the pandemic.

### 2.2. Variables

In this study, the dependent variable was panic selling during the COVID-19 pandemic. This was evaluated by responses to the question, “How did you modify your investment position in March 2020 when the stock market plummeted due to the spread of the new coronavirus?” The respondents were presented with five response options: (1) I sold some of my stocks/investment trusts, (2) I sold all my stocks/investment trusts, (3) I increased the amount of stocks and investment trusts, (4) I purchased new stocks/investment trusts, and (5) there were no particular changes to the amount invested in stocks/investment trusts. According to the research objective, the complete liquidation of stocks was primarily considered as an indicator of panic selling. Consequently, a binary dependent variable, “Panic selling_all”, was established, which was set to 1 if the respondent sold all their stocks and investment trusts, and 0 otherwise. To verify the reliability of the findings, a more lenient measure of panic selling was employed, which included both complete and partial selling of stocks. Therefore, a new binary variable, “Panic selling_all or partial”, was introduced, which was set to 1 if the respondents sold all or a portion of their stocks and investment trusts and 0 otherwise.

The primary independent variable in our study was hyperbolic discounting, which we estimated using two carefully designed survey questions. These questions were aimed at capturing respondents’ tendencies to prefer immediate rewards over delayed ones, which is a core characteristic of hyperbolic discounting. The first question presented respondents with a series of eight scenarios in which they had to choose between receiving a certain amount of money after a short delay of 2 days or a slightly longer delay of 9 days. The amount of money offered for the longer delay varied across these scenarios, allowing us to observe the point at which respondents switched their preference from the sooner reward to the later one. The second question followed a similar structure but involved a much longer delay, ranging from 90 to 97 days. Respondents were again asked to choose between receiving a smaller amount of money sooner or a larger amount later. By comparing the choices made in these two sets of scenarios, we calculated two discount rates: DR1, derived from the short-term (2 to 9 days) choices, and DR2, derived from the long-term (90 to 97 days) choices.

The concept of hyperbolic discounting suggests that individuals are likely to apply a higher discount rate to rewards in the near term, reflecting a stronger preference for immediate gratification. Therefore, if DR1 was greater than DR2, it indicated a tendency toward hyperbolic discounting, which we captured with a binary variable set to 1. If the DR1 was not greater than DR2, indicating a more consistent or less impulsive preference across time, the variable was set to 0. This method of measuring hyperbolic discounting follows established practices in behavioral economics, where a higher discount rate for immediate rewards is seen as indicative of a higher likelihood of making impulsive financial decisions. A detailed explanation of the survey questions and the calculation process for DR1 and DR2, along with how they inform the hyperbolic discounting variable, is provided in Appendix A.

Finally, we considered various demographic and socioeconomic factors to determine the unique effect of hyperbolic discounting on panic selling behavior. We incorporated several control variables, including gender, age, marital status, the presence of children, employment status, education level, urban residency, household income, household assets, financial literacy, risk aversion, overconfidence in financial knowledge, and a myopic view of the future. While most of the variables are self-explanatory, the measurement of financial literacy requires an explanation. We utilized the three-question methodology from Lusardi and Mitchell [26], as detailed in Appendix B, to measure financial literacy. These questions assess an individual’s mathematical ability and understanding of fundamental financial concepts, including interest rates, inflation, and risk diversification—key factors in making informed investment decisions. Each correct answer is assigned a score of 1, while incorrect answers receive a score of 0. The financial literacy variable is then calculated by averaging the scores of the three questions with equal weight given to each. Table 1 provides the comprehensive definitions and measurements of the dependent, independent, and control variables.

### 2.3. Descriptive Statistics

Table 2 provides a comprehensive statistical overview of the collected data. The dependent variables indicated the proportion of respondents who sold their stocks during the study period. Specifically, only 1.05% of the respondents sold all their stocks, while 3.87% of the respondents sold only a part, and a slightly larger proportion (4.91%) sold either a portion or all stocks. The primary independent variable, hyperbolic discounting, was exhibited by approximately 11.24% of respondents. This behavior is characterized by a preference for immediate, smaller rewards over delayed, larger rewards

Regarding demographic and socioeconomic variables, the average respondent was 43.71 years old with males constituting the majority (61.02%). A significant proportion of the respondents was married (65.93%), and most had children (56.94%). Education level was high among the respondents with 65.93% possessing a university degree. The unemployment rate was relatively low (5.11%). Urban residency was low (16.07%). Although this may seem unusual, the low urban residency rate likely reflects the broader geographic diversity of the survey’s target population, which includes active securities account holders from various regions, not just urban centers. Financial literacy was high among the respondents with an average score of 0.7986 of 1. The reported average household income and assets were JPY 7,502,653 and JPY 18,900,000, respectively. Lastly, psychological variables revealed that most respondents (53.60%) exhibited risk-averse behavior, and approximately 6.5% were overconfident in their financial knowledge. Meanwhile, 14.66% of the participants demonstrated a myopic view of the future, indicating a propensity to focus on short-term results.

Table 3 shows the association between hyperbolic discounting behavior and the act of selling all stocks and mutual funds. We use *t*-tests to compare panic selling among hyperbolic and nonhyperbolic discounters. Specifically, individuals who exhibit a complete sell-off of stocks and funds are more likely to display pronounced hyperbolic discounting tendencies than those who do not.

### 2.4. Methods

This study explores the role of hyperbolic discounting in the decision-making process that affects panic selling during a pandemic. This component of time discounting is crucial for understanding maladaptive decisions. Our central hypothesis posits that individuals prone to panic selling are likely to demonstrate higher discount rates, which is indicative of irrational and impulsive behavior. This perspective aligns with financial behavioral theories, suggesting that an individual lacking perseverance would opt for an immediate reward, thus subjectively discounting the value of a larger future reward [27]. To quantify this behavior, we employed two mathematical models. The first model posits that an outcome with a utility A if received immediately (*t* = 0) is valued at $A\cdot \delta^{t}$ if it is timed *t* periods into the future. Thus, the present time value (*V*) of receiving (*A*) at time (*t*) is as follows:(1)$$V(A,t)=A\cdot \delta^{t}$$
where the discount rate *δ* signifies the fixed proportional decrease in value for each added period. However, empirical observations suggest that humans tend to violate the exponential assumption of a constant proportional discount factor per unit time. They tend to discount rewards that appear in the immediate future more sharply than those that appear in the distant future [28]. To account for this discrepancy, we introduce a second model, the hyperbolic discount function, which provides a more accurate representation of the discount function estimated from the observed choices. This acts as an interpreted description of time discounting:(2)$$V(A,t)=\;A\cdot \frac{1}{1+k\cdot t}$$
where *k* represents the hyperbolic discount rate, capturing the degree of impulsivity in an individual’s behavior.

After establishing the hyperbolic function to capture the decision-making process behind panic selling, we performed a probit regression to test our hypothesis given that our dependent variable was binary. The combined effects of the main independent variable (hyperbolic discounting) and panic selling behavior are as follows:(3)$$Y_{i}\;=f({\;HD}_{i},X_{i},\varepsilon_{i})$$
where $Y_{i}$ is the measure of the dependent variable, “Panic_selling_all;” ${HD}_{i}$ represents hyperbolic discounting variables; $X_{i}$ is a vector of respondents’ various socioeconomic, demographic, and psychological variables; and $\varepsilon_{i}$ is the error term. A similar equation was used as an alternative measure of panic selling: panic selling_all or partial.

To avoid potential intercorrelation issues among the independent variables, we conducted correlation and multicollinearity tests for all the models. Our findings showed a weak association between the explanatory variables (<0.7) and no multicollinearity in all the models (variance inflation factor < 3). The test results are available upon request. This comprehensive analysis provides a better understanding of the role of hyperbolic discounting in panic selling behavior during a pandemic. The specifications of Equation (3) for full liquidation, full or partial liquidation, and only partial liquidation are as follows:(4)$${Panic\;selling\_all}_{i}=\beta_{0}+\beta_{1}{HD}_{i}+\beta_{2}{age}_{i}+\beta_{3}{age\;squared}_{i}+\beta_{4}{male}_{i}+\beta_{5}{edu}_{i}+\beta_{6}{UrbRes}_{i}+\;\beta_{7}{unemployment}_{i}+\beta_{8}{married}_{i}+\beta_{9}{children}_{i}+\beta_{10}{financial\;literacy}_{i}+\beta_{11}{HHIncome}_{i}+\;\beta_{12}{HHAssets}_{i}+\beta_{13}{risk\;aversion}_{i}+\beta_{14}{myopia}_{i}{+\beta_{15}{overconfidence}_{i}+\varepsilon }_{i}$$
(5)$$Panic\;selling\_\;all\;and\;{partial}_{i}=\beta_{0}+\beta_{1}{HD}_{i}+\beta_{2}{age}_{i}+\beta_{3}{age\;squared}_{i}+\beta_{4}{male}_{i}+\beta_{5}{edu}_{i}+\;\beta_{6}{UrbRes}_{i}+\beta_{7}{unemployment}_{i}+\beta_{8}{married}_{i}+\beta_{9}{children}_{i}+\beta_{10}{financial\;literacy}_{i}+\beta_{11}{HHIncome}_{i}+\;\beta_{12}{HHAssets}_{i}+\beta_{13}{risk\;aversion}_{i}+\beta_{14}{myopia}_{i}\;{+\beta_{15}{overconfidence}_{i}+\varepsilon }_{i}$$
(6)$$Panic\;selling\_\;{partial}_{i}=\beta_{0}+\beta_{1}{HD}_{i}+\beta_{2}{age}_{i}+\beta_{3}{age\;squared}_{i}+\beta_{4}{male}_{i}+\beta_{5}{edu}_{i}+\beta_{6}{UrbRes}_{i}+\;\beta_{7}{unemployment}_{i}+\beta_{8}{married}_{i}+\beta_{9}{children}_{i}+\beta_{10}{financial\;literacy}_{i}+\beta_{11}{HHIncome}_{i}+\;\beta_{12}{HHAssets}_{i}+\beta_{13}{risk\;aversion}_{i}+\beta_{14}{myopia}_{i}{+\beta_{15}{overconfidence}_{i}+\varepsilon }_{i}$$
where *HD* = hyperbolic discounting, *edu* = university degree, *UrbRes* = urban residency, *HHIncome* = ln of household income, *HHAssets* = ln of household assets, *myopia* = myopic view of the future, and *overconfidence* = overconfidence in financial literacy.

## 3. Estimation Results

To understand the relationship between panic selling and hyperbolic discounting, we conduct a cross-sectional probit regression analysis (Table 4). Initially, the binary dependent variable measures panic selling as the sale of all stocks and mutual funds. These findings indicate that hyperbolic discounting significantly predicts panic selling in all the models. This is evident from the positive coefficients significant at the 1% level, suggesting that investors with higher hyperbolic discounting levels are more inclined to sell their stocks and mutual funds. The positive association between hyperbolic discounting and panic selling underscores the impact of cognitive bias on investor behavior.

Regarding the demographic variables, male investors are more prone to panic selling, whereas those with university degrees are less likely to do so. Age shows a negative correlation with panic selling, which is significant at the 10% level, but only in Model 1.2. Other variables such as urban residency, unemployment, marital status, and having children do not exhibit a significant association with panic selling.

Considering socioeconomic variables, higher levels of financial literacy, increased household income, and assets are negatively associated with panic selling. By contrast, psychological variables such as risk aversion and a myopic view of the future demonstrate varying degrees of association with panic selling. Specifically, risk aversion shows a positive association with panic selling, whereas a myopic view of the future does not exhibit a significant association with panic selling. However, overconfidence in financial knowledge is positively associated with panic selling behavior.

To further validate our earlier findings, we conducted robustness tests to examine the relationship between hyperbolic discounting and panic selling behavior among investors. Table 5 and Table 6 present the estimation results when panic selling is measured as either a full or partial sale of stock and as a partial sale of stock, respectively. The results reaffirm the significance of hyperbolic discounting in predicting panic selling even when different measures of panic selling are considered. Consistent with our previous analysis, individuals who exhibit higher levels of hyperbolic discounting are more likely to engage in panic selling regardless of whether they sell all or only a portion of their stocks.

Concerning demographic factors, we find that certain characteristics remain influential in predicting panic selling behavior. Male investors are consistently more prone to panic selling, and the results are consistent in Table 5 and Table 6. However, the association of university degree varies with the measurements of panic selling employed in both tables. More specifically, respondents with at least a university degree are less likely to engage in either the partial or complete selling of stocks, although this effect diminishes in subsequent models. However, the direction of the variable’s coefficient under partial sale of stocks reverses. The significance of age and marriage is considerable with certain models showing a positive association under both robustness test results tables. This positive association of age with panic selling behavior could be due to the higher risk aversion of older people, shorter investment horizons as they approach retirement, past experiences with market downturns, immediate financial needs, and more diversified portfolios that allow for partial rather than complete sell-offs. Additionally, having children emerged as significant predictors of partial or full panic selling but did not significantly associate itself with the partial sell-off of stocks in this analysis. Unemployment is a significant predictor of different levels of panic selling behavior in both tables, indicating that unemployed individuals are more likely to engage in this behavior.

Moreover, several socioeconomic factors continue to influence panic selling behavior. Higher levels of financial literacy and household income are associated with a lower likelihood of different levels of panic selling behavior in both tables, whereas household assets become a positive moderator for the partial selling of stocks, but its presence is not persistent under the partial or complete selling of stocks. Psychological variables play a consistent role in both tables, with risk aversion positively linked to panic selling, suggesting that more risk-averse individuals may sell their stocks during times of market instability. A myopic view of the future becomes significantly positive with panic selling in this analysis under both types of measurement, while overconfident people consistently divert stocks impulsively regardless of the level of the stocks liquidated.

## 4. Discussion

Exploring how hyperbolic discounters react to panic selling is a crucial, yet underexplored, aspect of financial research, particularly during crises. Behavioral theories provide nuanced insights into how hyperbolic discounting can exacerbate panic selling tendencies by overemphasizing immediate loss aversions, leading to impulsive divestment from stocks [5,6,7,21]. Guided by the hypothesis that people exhibiting hyperbolic discounting are likely to engage in panic selling, our analysis provides insights into the intricate interplay between hyperbolic discounting and divestment behavior during the COVID-19 pandemic in Japan.

Our regression analysis shows that individuals demonstrating hyperbolic discounting tendencies are significantly more inclined to engage in panic selling behavior during crises. Furthermore, as part of robustness checks, we employed two alternative measures of panic selling behavior: (1) panic selling of all or partial stocks and (2) panic selling of partial stocks. Both alternative measures provide consistent results, reinforcing the association between hyperbolic discounting and increased likelihood of panic selling. This consistency across different measures confirms the robustness of our findings and supports the conclusion that individuals with hyperbolic discounting tendencies are more prone to panic selling in times of financial distress. This empirical validation substantiates the behavioral theories proposed by Kahneman and Tversky [21], Frederick et al. [7] and Loewenstein and Prelec [6], highlighting the pervasive influence of cognitive biases not only on investment in financial securities but also on the divestment of securities during market turmoil. This finding resonates with the broader context of our study, particularly within the Japanese economy, emphasizing the importance of understanding how hyperbolic discounting manifests loss aversion among investors in crises. During the initial months of the COVID-19 pandemic, its profound impact on financial markets induced panic selling among investors owing to fear of loss despite the relatively short duration of the crisis [29,30]. The Tokyo Stock Exchange experienced significant volatility and steep declines in stock prices, reflecting global trends amid the pandemic-induced uncertainty [31,32]. The increased fear of potential future declines in stock values and realization of substantial losses led to a surge in impulsive panic selling, exacerbating market turbulence and amplifying the challenges faced by Japanese investors [33,34]. This aligns with Barberis and Huang [5], who show that investors’ reactions to perceived threats to their financial well-being, driven by fear of loss, are often swift and emotional, leading to impulsive decision making. Our research illuminates the pivotal role of hyperbolic discounting in driving panic selling behaviors, offering valuable insights into investor decision making during crises.

In addition to focusing on the association between hyperbolic discounting and panic selling, we determine panic selling behavior based on other demographic factors such as age, gender, and educational status. Although the significance of age varied across models, age displayed a statistically significant negative association at the 10% level in Model 1.1, suggesting its potential impact on panic selling behavior. This observation corroborates the existing literature indicating the nuanced influence of age on risk-taking behavior [35]. Moreover, we observe that males are more likely to engage in panic selling, which can be explained by their lower tendency to take risks under unstable economic conditions during the pandemic. Although it is generally assumed that women are more risk averse than men [36], this gender stereotyping is challenged depending on circumstances and other aspects of investor preference [37]. During crises, men show a propensity for less risk tolerance, thus following herding behavior of selling all stocks and engaging in panic selling [37,38]. Moreover, our analysis reveals a negative association between higher educational levels (university degree and above) and panic selling behavior. This inverse relationship is attributed to the influence of higher education in fostering a more sophisticated understanding of market risks and improving emotional regulation. This reduces the propensity to panic sell during market downturns [39].

Among socioeconomic factors, variables such as financial literacy, household income, and assets show a significant association with panic selling behavior. Specifically, individuals with higher financial literacy diverge from panic selling tendencies. This is because financial literacy strengthens investors’ confidence and risk-taking capacity during economic instability, equipping them with comprehensive evaluations of real-time investment performance [40,41]. Moreover, our findings indicate that households with more assets are less inclined to engage in panic selling during the pandemic. This observation corroborates Bucher-Koenen and Ziegelmeyer [42], indicating that individuals with more wealth are less inclined to sell some or all their assets during financial crises, potentially because of their reduced vulnerability to income shocks [43]. Similarly, a higher household income reduces panic selling during the COVID-19 pandemic, consistent with Kato’s findings [44], because households with high incomes are more likely to benefit from professional advice that discourages panic selling [45,46].

Among the psychological variables, our results reveal that risk aversion increases the chance of panic selling during the pandemic. This finding is consistent with Wendel [47], who highlights how the risk preference approach can increase conservative investment among investors because of the fear of losses, effectuating panic selling during market downturns. Finally, we observe that investors displaying overconfidence in their financial knowledge are more prone to divest from stocks, which is consistent with prior research indicating that increased overconfidence may drive impulsive decisions during market instability [48,49].

Although this study provides valuable insights, it has some limitations. First, this study defines panic selling as the sale of all or a significant portion of holdings during the pandemic. However, this classification may not fully capture the nuances of investor behavior, as some sales may have been driven by factors other than panic. Additionally, due to the lack of data on whether these investors reentered the market, we cannot definitively confirm that these sales were made in a state of panic. Second, our data focused solely on internet-based customers, limiting the generalizability of our findings. Third, our analysis was based on panic selling data from a single wave, restricting the depth of longitudinal evidence available on panic selling behavior. Therefore, future research should employ broader data collection methods and longitudinal studies, exploring how panic selling behavior evolves among hyperbolic discounters as the pandemic situation improves gradually.

## 5. Conclusions

Drawing on a range of behavioral theories, this study is the first to present empirical evidence suggesting that people susceptible to hyperbolic discounting engage in panic selling behavior during crises. Specifically, it offers behavioral explanations for the observed panic selling behavior and suggests that impulsive investors may overemphasize aversion to immediate loss owing to fear during crises. This impulsivity-led fear of loss triggers investors to divest from stocks without fully considering the potential consequences, increasing market volatility. Among the control variables, certain demographic and socioeconomic characteristics, namely age, male gender, education level, financial literacy, household income and assets, risk aversion, and overconfidence in financial knowledge, significantly affected panic selling behavior. This accentuates the multifaceted nature of investors’ decision-making processes, highlighting the importance of considering a comprehensive range of control variables to effectively understand and address panic selling behavior.

The findings of this study have important implications for practitioners and policymakers. Our study highlights the importance of addressing cognitive biases in financial decision making, particularly during market crises. Strategies such as targeted financial education programs, regulatory interventions to prevent market manipulation, and the provision of professional advice to investors can mitigate the impact of hyperbolic discounting and panic selling. These interventions can stabilize financial markets and protect investors’ long-term interests by promoting awareness regarding behavioral biases and encouraging rational decision making.

## Figures and Tables

**Figure 1 behavsci-14-00795-f001:**
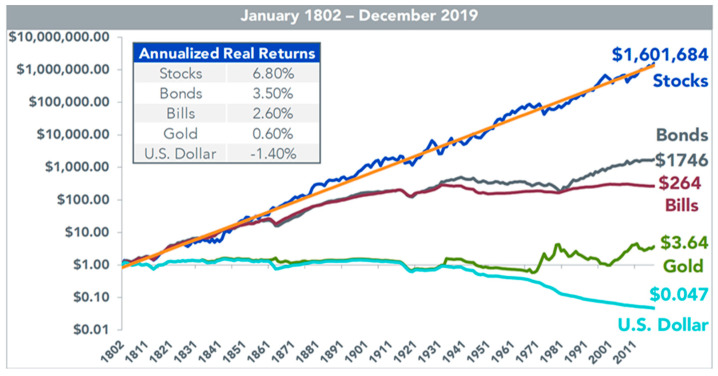
Total return indices for the last 200 years. Historical data of long-term cumulative growth of $1 from major asset classes along with the average annual real returns accumulating from those long-term wealth gains from 1802 to 2019. The figure is adopted from Siegel [10].

**Figure 2 behavsci-14-00795-f002:**
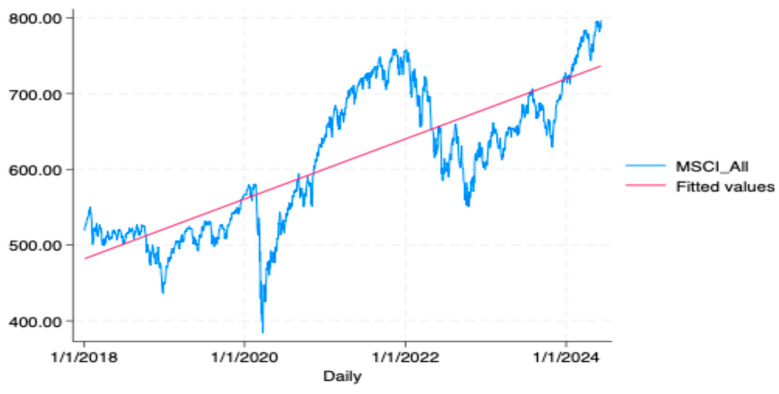
Global market decline during March 2020 due to COVID-19 pandemic. The data visualize the sharp fall in global market indices in March 2020, which was a direct result of the COVID-19 pandemic. The steep decline illustrates the immediate economic impact and investor reaction to the global health crisis. However, it also demonstrates the gradual recovery observed in the months following the initial decline. The data are sourced from the MSCI World Index.

**Table 1 behavsci-14-00795-t001:** Variable definitions.

Dependent Variable	
Panic selling_all	Binary variable: 1 if a respondent sold all stocks and mutual funds, and 0 otherwise
Panic selling_all or partial	Binary variable: 1 if a respondent sold part of or all stocks and mutual funds, and 0 otherwise
Panic selling_partial	Binary variable: 1 if a respondent sold only part of their stocks and mutual funds, and 0 otherwise
Independent variables	
Hyperbolic Discounting	Binary variable: 1 if DR1 > DR2, and 0 otherwise
Age	Continuous variable: respondent’s age
Age Squared	Continuous variables: square of the respondent’s age
Male	Binary variable: 1 if the respondent is a male
University Degree	Binary variable: 1 if the respondent holds at least a university degree, and 0 otherwise
Urban Residency	Binary variable: 1 if the respondent resides in an urban area, and 0 otherwise
Unemployment	Binary variable: 1 if the respondent is unemployed
Married	Binary variable: 1 if the respondent is married, and 0 otherwise
Have Children	Binary variable: 1 if the respondent has at least one child, and 0 otherwise
Financial Literacy	Continuous variable: average score for the number of correct answers from the three financial literacy questions
Household Income	Continuous variable: the respondent’s income measured in Japanese yen
Log of Household Income	Continuous variable: natural log of the respondent’s personal income
Household Asset	Continuous variable: household financial assets measured in Japanese yen
Log of Household Asset	Continuous variable: natural log of the respondent’s household financial assets
Risk Aversion	Continuous variable: risk of rain preference (percentage score from the question, “Usually when you go out, how high must the probability of rainfall be before you take an umbrella?”)
Overconfidence in Financial Knowledge	Binary variable: 1 if respondent scores below average in financial literacy questions but confident about his/her financial knowledge, and 0 otherwise
Myopic View of the Future	Binary variable: 1 if the respondent agrees or completely agrees with the statement, “Since the future is uncertain, it is a waste to think about it”, and 0 otherwise

**Table 2 behavsci-14-00795-t002:** Descriptive statistics.

Variable	Mean	Std. Dev.	Min	Max
Dependent Variables				
Panic selling_all				
Otherwise (0)	0.9895122	0.101872	0	1
Sold all (1)	0.0104878	0.101872	0	1
Panic selling_partial				
Otherwise (0)	0.9613978	0.192646	0	1
Sold partial (1)	0.0386022	0.192646	0	1
Panic selling_all or partial				
Otherwise (0)	0.9509099	0.216057	0	1
Sold partial or whole (1)	0.0490901	0.216057	0	1
Main Independent Variable				
Hyperbolic Discounting	0.1124113	0.315873	0	1
Other Independent Variables				
Age	43.71476	11.70996	18	94
Age Squared	2048.103	1081.685	324	8836
Male	0.61018	0.487711	0	1
University Degree	0.6592855	0.473951	0	1
Urban Residency	0.1607434	0.367296	0	1
Unemployment	0.0510778	0.220157	0	1
Married	0.6579629	0.474394	0	1
Have Children	0.5693889	0.495164	0	1
Financial Literacy	0.7986228	0.294304	0	1
Household Income	7,502,653	4,119,226	1,000,000	20,000,000
Log of Household Income	15.66981	0.600837	13.81551	16.81124
Household Asset	18,900,000	23,400,000	2,500,000	100,000,000
Log of Household Asset	16.15078	1.076472	14.7318	18.42068
Risk Aversion	0.535954	0.228671	0	1
Overconfidence in Financial Knowledge	0.0651234	0.2467444	0	1
Myopic View of the Future	0.146605	0.353713	0	1
Observations		129,293		

**Table 3 behavsci-14-00795-t003:** Panic selling by hyperbolic discounting.

	Hyperbolic Discounting		
Sold All Stocks	0	1	Total
0	113,640	14,297	127,937
%	99.02	98.37	99
1	1119	237	1356
%	0.98	1.63	1
Total	114,759	14,534	129,293
%	100	100	100
Mean Difference	*t*-value = −7.3106 ***		

*** indicates *p* < 0.01.

**Table 4 behavsci-14-00795-t004:** Estimation results of panic selling of all stocks.

Variables	Model 1.1	Model 1.2	Model 1.3	Model 1.4
	Dependent Variable: Sold All Stocks			
Hyperbolic Discounting	0.1973 ***	0.1811 ***	0.1646 ***	0.1636 ***
	(0.0282)	(0.0285)	(0.0288)	(0.0289)
Age		−0.0102 *	−0.0012	−0.0004
		(0.0061)	(0.0063)	(0.0063)
Age Squared		0.0001	0.0000	0.0000
		(0.0001)	(0.0001)	(0.0001)
Male		0.3796 ***	0.4591 ***	0.4544 ***
		(0.0252)	(0.0270)	(0.0269)
University Degree		−0.1606 ***	−0.0758 ***	−0.0814 ***
		(0.0220)	(0.0235)	(0.0236)
Urban Residency		0.0330	0.0438	0.0415
		(0.0281)	(0.0285)	(0.0286)
Unemployment		0.0535	0.0476	0.0494
		(0.0490)	(0.0527)	(0.0527)
Married		0.0039	0.0349	0.0362
		(0.0286)	(0.0301)	(0.0302)
Have Children		−0.0361	−0.0438	−0.0419
		(0.0284)	(0.0286)	(0.0287)
Financial Literacy			−0.5153 ***	−0.5094 ***
			(0.0346)	(0.0345)
Log of Household Income			−0.0435 *	−0.0418 *
			(0.0229)	(0.0229)
Log of Household Asset			−0.0365 ***	−0.0399 ***
			(0.0123)	(0.0123)
Risk Aversion				0.2116 ***
				(0.0472)
Myopic View of the Future				0.0126
				(0.0297)
Overconfidence in Financial Knowledge				0.1335 ***
				(0.0360)
Constant	−2.3545 ***	−2.2760 ***	−0.9961 ***	−1.0938 ***
	(0.0116)	(0.1359)	(0.3369)	(0.3365)
Observations	129,293	129,293	129,293	129,293
Log Likelihood	−7486	−7339	−7201	−7190
Chi^2^ Statistics	94.88	351.9	574.8	581.1
*p*-value	0	0	0	0

Robust standard errors in parentheses. *** *p* < 0.01, and * *p* < 0.1.

**Table 5 behavsci-14-00795-t005:** Estimation results of panic selling of all or partial stocks.

Variables	Model 1.1	Model 1.2	Model 1.3	Model 1.4
	Dependent Variable: Sold Partial or All Stocks			
Hyperbolic Discounting	0.1560 ***	0.1338 ***	0.1254 ***	0.1248 ***
	(0.0174)	(0.0176)	(0.0177)	(0.0177)
Age		0.0016	0.0074 **	0.0079 **
		(0.0035)	(0.0036)	(0.0036)
Age Squared		0.0000	−0.0000	−0.0000
		(0.0000)	(0.0000)	(0.0000)
Male		0.3624 ***	0.4114 ***	0.4096 ***
		(0.0140)	(0.0145)	(0.0145)
University Degree		−0.0384 ***	0.0077	0.0040
		(0.0131)	(0.0137)	(0.0137)
Urban Residency		0.0137	0.0229	0.0212
		(0.0164)	(0.0166)	(0.0166)
Unemployment		0.1129 ***	0.0803 ***	0.0816 ***
		(0.0270)	(0.0286)	(0.0285)
Married		0.0111	0.0394 **	0.0405 **
		(0.0163)	(0.0173)	(0.0173)
Have Children		−0.0324 **	−0.0335 **	−0.0316 *
		(0.0161)	(0.0162)	(0.0162)
Financial Literacy			−0.3494 ***	−0.3468 ***
			(0.0214)	(0.0214)
Log of Household Income			−0.0606 ***	−0.0598 ***
			(0.0129)	(0.0129)
Log of Household Asset			0.0105	0.0088
			(0.0069)	(0.0069)
Risk Aversion				0.1374 ***
				(0.0268)
Myopic View of the Future				0.0336 **
				(0.0171)
Overconfidence in Financial Knowledge				0.1572 ***
				(0.0218)
Constant	−1.6931 ***	−2.0842 ***	−1.2496 ***	−1.3189 ***
	(0.0066)	(0.0798)	(0.1899)	(0.1902)
Observations	129,293	129,293	129,293	129,293
Log Likelihood	−25,212	−24,664	−24,512	−24,497
Chi^2^ Statistics	224.4	1255	1526	1555
*p*-value	0	0	0	0

Robust standard errors in parentheses. *** *p* < 0.01, ** *p* < 0.05, and * *p* < 0.1.

**Table 6 behavsci-14-00795-t006:** Estimation results of panic selling of partial stocks.

Variables	Model 1.1	Model 1.2	Model 1.3	Model 1.4
	Dependent Variable: Sold Partial Stocks			
Hyperbolic Discounting	0.1236 ***	0.1003 ***	0.0951 ***	0.0937 ***
	(0.0190)	(0.0193)	(0.0193)	(0.0193)
Age		0.0051	0.0088 **	0.0102 ***
		(0.0038)	(0.0039)	(0.0039)
Age Squared		0.0000	−0.0000	−0.0000
		(0.0000)	(0.0000)	(0.0000)
Male		0.3217 ***	0.3564 ***	0.3485 ***
		(0.0150)	(0.0155)	(0.0155)
University Degree		0.0108	0.0370 **	0.0334 **
		(0.0142)	(0.0149)	(0.0149)
Urban Residency		0.0060	0.0122	0.0109
		(0.0178)	(0.0179)	(0.0179)
Unemployment		0.1205 ***	0.0829 ***	0.0812 ***
		(0.0286)	(0.0302)	(0.0302)
Married		0.0134	0.0354 *	0.0371 **
		(0.0177)	(0.0187)	(0.0187)
Have Children		−0.0274	−0.0256	−0.0240
		(0.0174)	(0.0174)	(0.0174)
Financial Literacy			−0.2606 ***	−0.2358 ***
			(0.0232)	(0.0236)
Log of Household Income			−0.0562 ***	−0.0573 ***
			(0.0138)	(0.0138)
Log of Household Asset			0.0273 ***	0.0216 ***
			(0.0073)	(0.0073)
Risk Aversion				0.0915 ***
				(0.0288)
Myopic View of the Future				0.0368 **
				(0.0185)
Overconfidence in Financial Knowledge				0.1450 ***
				(0.0236)
Constant	−1.7824 ***	−2.2974 ***	−1.7849 ***	−1.7817 ***
	(0.0069)	(0.0867)	(0.2042)	(0.2045)
Observations	129,293	129,293	129,293	129,293
Log Likelihood	−21,116	−20,660	−20,587	−20,562
Chi^2^ Statistics	42.41	919.3	1073	1128
*p*-value	7.38 e−11	0	0	0

Robust standard errors in parentheses. *** *p* < 0.01, ** *p* < 0.05, and * *p* < 0.1.

## Data Availability

The data that support the findings of this study were collected by Rakuten Securities in collaboration with Hiroshima University. These data are not publicly available due to restrictions under the licensing agreement for the current study. However, they can be made available from the authors upon reasonable request and with permission from Rakuten Securities and Hiroshima University.

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
