# Peer review of "Unraveling Investor Behavior: The Role of Hyperbolic Discounting in Panic Selling Behavior on the Global COVID-19 Financial Crisis"

_behavsci, 2024, doi:10.3390/bs14090795_

Round 1
Reviewer 1 Report
Comments and Suggestions for Authors
Please see the report.

Reviewer 2 Report
Comments and Suggestions for Authors
Interesting paper. The authors analyze the issue of panic selling or more in general overreactions to the market based on philological biases rather than on fundamentals. There is a large body of literature covering this type of topic but it is also a large and complex field so more papers like this one are needed. A core part of this paper is the questionnaire, which is always a complex and time consuming task.
I have a few comments
1) The questionnaire is a key part of this paper and the authors describe it but I think that more detail should be provided. For example, in the sentence “After excluding incomplete data, the final sample size was reduced to 129,293 observations, representing 68.22% of 189,524 valid responses.” I think that it not clear enough what data was excluded. It will be important to describe a series of objective criteria for inclusion/exclusion, particularly because it seems that a relatively large percentage was excluded.
2) In line 202 it is mentioned that urban residency was low (~16%). This seems unusual and requires some explanation as it could (in principle) skew the data. I think that a qualitative explanation of why this is the case would suffice.
3) In equations 4 and 5 I would suggest using abbreviations for the terms (and then defining them). It is a bit hard to follow the equations when presented like this.
4) I think that it would be interesting providing some more insight into the results. For example, in line 305 it is mentioned “The significance of age is considerable, with certain models showing a positive association.”… Any potential explanation on why this could be the case? Whtas the view from the authors?
5) The presentation of some of the tables and graphs could be improved. Sorry if I miss it but what’s the red circle in Figure 2. Maybe I missed the explanation.
Comments on the Quality of English LanguageSuggest polishing it a bit.
Round 2
Reviewer 1 Report
Comments and Suggestions for Authors
The authors have done a good job addressing my concerns and comments. Overall, I think the quality of the paper has improved.